# Three Dimensional Shape Reconstruction via Polarization Imaging and Deep Learning

**DOI:** 10.3390/s23104592

**Published:** 2023-05-09

**Authors:** Xianyu Wu, Penghao Li, Xin Zhang, Jiangtao Chen, Feng Huang

**Affiliations:** School of Mechanical Engineering and Automation, Fuzhou University, Fuzhou 350108, China; xwu@fzu.edu.cn (X.W.);

**Keywords:** deep learning, polarization imaging, shape from polarization, surface normal estimation

## Abstract

Deep-learning-based polarization 3D imaging techniques, which train networks in a data-driven manner, are capable of estimating a target’s surface normal distribution under passive lighting conditions. However, existing methods have limitations in restoring target texture details and accurately estimating surface normals. Information loss can occur in the fine-textured areas of the target during the reconstruction process, which can result in inaccurate normal estimation and reduce the overall reconstruction accuracy. The proposed method enables extraction of more comprehensive information, mitigates the loss of texture information during object reconstruction, enhances the accuracy of surface normal estimation, and facilitates more comprehensive and precise reconstruction of objects. The proposed networks optimize the polarization representation input by utilizing the Stokes-vector-based parameter, in addition to separated specular and diffuse reflection components. This approach reduces the impact of background noise, extracts more relevant polarization features of the target, and provides more accurate cues for restoration of surface normals. Experiments are performed using both the DeepSfP dataset and newly collected data. The results show that the proposed model can provide more accurate surface normal estimates. Compared to the UNet architecture-based method, the mean angular error is reduced by 19%, calculation time is reduced by 62%, and the model size is reduced by 11%.

## 1. Introduction

Existing 3D imaging technologies, such as binocular stereo vision, primarily use the intensity and texture information of light reflected by an object’s surface to reconstruct its 3D shape [1,2,3]. Smooth and highly reflective objects with low surface texture features are prone to overexposure, which can result in significant information loss in the reconstruction output of binocular stereo vision. Structured light 3D imaging may also suffer from large areas of overexposure that can impede accurate extraction of grating stripes, leading to unreliable depth information. As a consequence, these techniques may be challenging to apply when reconstructing the shape of smooth and non-textured objects due to missing data in the imaging results. To overcome these limitations, polarization imaging technology is increasingly used in the field of 3D imaging [4,5]. Polarization 3D imaging methods, also known as the shape from polarization (SfP) method, utilizes the polarization information of reflected light to perform shape estimation and effectively reduces the impact of surface texture loss and poor lighting conditions [6]. The key steps of SfP include analyzing the polarization characteristics of reflected light [7] and establishing a mathematical relationship between the polarization information and the normal direction of the object’s surface micro-facet to restore the shape of the target surface. SfP methods have the advantages of high precision, long working distance, non-contact, and strong anti-interference ability. The polarization characteristics of the light reflected from the target can be obtained even under poor illumination conditions.

The use of a Cartesian coordinate system in the image plane allows for the establishment of a geometric relationship model between the surface normal $\vec{n}$ of an object and its zenith angle θ and azimuth angle φ (Figure 1). This enables the determination of the surface normal distribution. However, two surfaces with different orientations (i.e., the values of zenith angle and azimuth angle) can produce exactly the same polarization characteristics, making it difficult for the polarization 3D imaging technology to solve the multivalued problem of zenith angle and azimuth angle. Researchers have proposed various solutions to eliminate the ambiguity of surface normals [8,9,10,11,12,13,14], with the core step being the establishment of the relationship between the polarization characteristics of reflected light and normal direction of the target surface based on Fresnel’s law and combining other means to solve the uncertainty in the 3D reconstruction process [8,10,11,12,13]. Koshikawa [15] calculated the target surface normal by analyzing the change in the polarization state of the circularly polarized light reflected by the surface of the insulator. Miyazaki [8] combined the advantages of polarization imaging and the space carving technique, proposing a polarization 3D imaging technology based on multiview observation to achieve 3D imaging of black highly reflective targets. Smith [9] proposed a surface linear depth estimation method based on sparse linear equations to solve azimuth ambiguity, allowing for the direct estimation of surface depth from a single polarization image. Mahmoud [10] proposed a polarization 3D imaging method that combined the shadow recovery method to eliminate azimuth ambiguity and obtain the surface normal of an object from a spectral image. Kadambi et al. [11,12] proposed a method that fuses the depth map acquired by Kinect with polarized 3D imaging to achieve high-precision and robust polarized 3D imaging under both indoor and outdoor lighting conditions. Cui et al. [13] proposed a polarization 3D imaging method that combined multiview observation with a new polarization imaging model, which is different from the previous SfP model that was limited to diffuse reflection or specular reflection, and can handle real-world objects with mixed specular and diffuse polarization information. To utilize polarization information for estimating the relative pose between two views, Cui et al. [14] proposed a relative pose estimation method from polarized images using a minimal solver. This method is suitable for objects dominated by specular or diffuse reflections. The aforementioned studies analyzed the multivalued problem of incident and azimuth angle in the polarized 3D imaging process of specular and diffuse reflected light and proposed methods to solve this problem. The reflection light of natural objects typically contains both specular and diffuse components; however, the existing polarization-based 3D imaging techniques that typically only consider one component for surface normal estimation. These techniques have limitations, with specular-based methods unable to measure the uniqueness of the incident angle, while diffuse-based methods require additional 3D perception technologies for azimuth angle information. Moreover, these methods have limited material applicability and often require computationally complex disambiguation processes.

Recently, researchers have been applying deep learning methods in the field of polarization 3D imaging [16,17,18,19,20]. Ba et al. [21] proposed using a deep learning network to address challenges encountered during the calculation and reconstruction process of polarization 3D imaging. The trained model provided robust surface normal estimation results under three different types of indoor and outdoor illumination conditions. Kondo et al. [22] proposed a new pBRDF model to describe the polarization properties of various materials and created a renderer to generate a large number of realistic polarization images for training a polarization 3D imaging model. Deschaintre et al. [23] proposed a method for effectively acquiring high-quality shape estimation and spatially varying reflectivity of 3D objects by combining polarization information with deep learning under flash lighting. Lei et al. [24] proposed a new data-driven polarization 3D imaging method based on physics prior for scene-level normal estimation from polarization images. The aforementioned methods were proposed based on the UNet architecture for estimating the target surface normal [25]. The UNet framework is an encoder–decoder architecture, where the encoder is designed for feature extraction and the decoder is designed for upsampling and restoration. The encoder module initially focuses on extracting detailed textures and local features. As the network downsamples through convolution, the encoder’s receptive field increases, allowing the network to extract the global features of the input. During the upsampling process, feature channels are designed in the encoder to preserve as many detailed features as possible to retrieve the lost context information.

In the field of SfP, the UNet-based network [21] used for polarization 3D imaging tends to lose information in the fine texture area of the target, leading to inaccurate normal estimation and reduced reconstruction accuracy. Additionally, it has limitations in recovering target texture details and surface normal estimation accuracy. To overcome these limitations, a U^2^Net architecture-based polarization 3D imaging network is proposed. Our proposed method offers the following advantages:Our method enhances the reconstruction of object textures by extracting more comprehensive information, mitigating the loss of texture information during the process.By reducing the number of parameters and increasing the network’s depth, our method enhances the model’s expressive and generalization abilities and reduces computational cost.We also improved the polarization representation of the input, which was considered to be more effective than the polarization representation used in other polarization 3D imaging methods [21,24], through qualitative and quantitative evaluations.

The proposed network was evaluated and validated using the DeepSfP dataset and experiments. Furthermore, our method achieved a significantly lower mean angular error (MAE) in estimating the surface normal of the target compared to the methods presented in [21,24].

## 2. Related Work

### 2.1. Shape from Polarization Principle

The 3D contour of a target object can be reconstructed by determining the normal of its surface microfacet, as outlined in [26]. Figure 2 illustrates the relationship between the normal direction of an object’s microfacet and the polarization characteristics of the reflected light beam [8]. This relationship can be mathematically expressed as [21].
(1)$$\vec{n}=\left(\cos\varphi \;\sin\theta ,\sin\varphi \;\sin\theta ,\cos\theta \right)$$

In order to solve for the microfacet normal vector, it is necessary to calculate the polarization characteristic parameters, including the zenith angle θ and azimuth angle φ.

In real-world scenarios, reflected light is typically partially polarized. Only when light is incident at Brewster angle, a complete linear polarization can be achieved [27]. The Fresnel equation states that while the diffuse reflection component $I_{d}$ remains constant for different polarization angles, the specular reflection component $I_{s}$ varies [28]. As the specular reflection is only partially polarized, a fraction of the specular reflection component $I_{s}$ is also constant, as shown in Figure 3. As per Malus’ Law, the intensity of light varies with the rotation of the polarizer. For different polarization directions $\phi_{pol}$, the specular reflection component $I_{s}$ can be represented as the sum of a constant component $I_{sc}$ and a cosine function term with an amplitude $I_{sv}$.

Images with different polarization angles can be captured using a polarization camera or by placing a rotating polarizer in front of a standard camera lens. The relationship between the observed light intensity $I\left(\phi_{pol}\right)$ and the polarization angle $\phi_{pol}$ can be expressed as [29]:(2)$$I\left(\phi_{pol}\right)=\frac{I_{max}+I_{min}}{2}+\frac{I_{max}-I_{min}}{2}\cos(2\left(\phi_{pol}-\phi \right))\;=I_{d}+I_{sc}+I_{sv}\cos2\left(\phi_{pol}-\phi \right)\;=I_{c}+I_{sv}\cos2\left(\phi_{pol}-\phi \right)$$
where $I_{max}$ and $I_{min}$ represent the upper and lower bounds of $I\left(\phi_{pol}\right)$, respectively. $I_{c}$ denote the constant reflection component, which is the sum of the diffuse reflection component $I_{d}$ and unpolarized specular reflection component $I_{sc}$. Due to the π ambiguity of the phase angle, ϕ and ϕ+180° produce the same light intensity in the captured images.

In general, a polarization camera can capture polarization images at four angles $\{\phi_{pol}\in \left(0{}^{\circ},\;45{}^{\circ},\;90{}^{\circ},\;\mathrm{and}\;135{}^{\circ}\right)$} in a single shot. By using these images, Formula (2) can be rewritten as:(3)$$P=\left[\begin{array}{c} 1,\;\cos2\phi_{0},\;\sin2\phi_{0}\\ 1,\;\cos2\phi_{45},\;\sin2\phi_{45}\\ 1,\;\cos2\phi_{90},\;\sin2\phi_{90}\\ 1,\;\cos2\phi_{135},\;\sin2\phi_{135} \end{array}\right]=\left[\begin{array}{c} 1\;\;\;\;\;\;\;\;1\;\;\;\;\;\;0\\ 1\;\;\;\;\;\;\;\;0\;\;\;\;\;\;1\\ 1\;\;-1\;\;\;\;\;\;0\\ \;\;1\;\;\;\;\;\;\;\;0\;\;-1 \end{array}\right]$$
(4)$$O=\left[\begin{array}{c} I_{c}\\ I_{sv}\cos2\phi \\ I_{sv}\sin2\phi \end{array}\right]$$
(5)$$I\left(\phi_{pol}\right)=\left[\begin{array}{c} I_{0}\\ I_{45}\\ I_{90}\\ I_{135} \end{array}\right]=PO=\left[\begin{array}{c} I_{c}+I_{sv}\cos2\left(\phi_{0}-\phi \right)\\ I_{c}+I_{sv}\cos2\left(\phi_{45}-\phi \right)\\ \begin{array}{c} I_{c}+I_{sv}\cos2\left(\phi_{90}-\phi \right)\\ I_{c}+I_{sv}\cos2\left(\phi_{135}-\phi \right) \end{array} \end{array}\right]$$
where the rotation angle of the polarizer allows to determine matrix P. $I_{0}$, $I_{45}$, $I_{90},$ and $I_{135}$ represent polarization images captured at the angles of 0°, 45°, 90° and 135°. By solving Equation (5), it is possible to calculate $I_{c}$ and $I_{sv}$ [30]. However, separating $I_{d}$ from $I_{c}$ remains a challenging problem. We can calculate the approximate specular reflection and diffuse reflection components:(6)$$I_{rawS}=2\times I_{sv}$$
(7)$$I_{rawD}=I_{c}-I_{rawS}$$
where $I_{rawS}$ and $I_{rawD}$ refer to the approximate specular reflection and diffuse reflection components proposed in [29].

### 2.2. Network Input

The polarization properties of materials are a valuable source of information for tasks such as shape estimation, material classification, and reflection component separation and can be effectively utilized by deep learning networks. Kondo et al. [22] used the polarization angle ϕ, modified degree of polarization ρ, and unpolarized intensity $I_{un}$ as inputs for the deep learning network. In this paper a new SfP network input is proposed, which includes the following polarization images: $I_{rawS}$, $I_{rawD}$, $\phi_{e}$, and $I_{\rho }$. Our proposed input provides a network with separated specular and diffuse information and can lead to an improved network performance compared to other methods.

$\phi_{e}$ denotes the encoded phase angle:(8)$$\phi_{e}=\left(\cos2\phi ,\sin2\phi \right)$$

$\phi_{e}$ is a vector, which aims to address the shortcomings of phase angle [24]. As in coded space, ϕ and ϕ+π represent the same angle.

$I_{\rho }$ is the Stokes-vector-based parameter [31], i.e., the degree of polarization
(9)$$\rho =\frac{\sqrt{S_{1}^{2}+S_{2}^{2}}}{S_{0}}$$
is modified to be
(10)$$I_{\rho }=\sqrt{S_{1}^{2}+{\left(S_{1}+S_{2}\right)}^{2}}$$
where $S_{0}$ represents the intensity image, which can be regarded as the sum of the light intensity in the horizontal direction and the vertical direction. $S_{1}$ represents the light intensity difference between the horizontal polarization state and the vertical polarization state. $S_{2}$ represents the light intensity difference between the 45° direction polarization state and the 135° direction polarization state.

The average contrast between the target and background in the $S_{1}$ parameter image is higher, indicating that it has more distinct target polarization characteristics. Equation (9) includes rich details and texture information in its denominator $S_{0}$, but it also introduces background noise. In comparison, the contrast between the target and background in the $I_{\rho }$ image is further improved, with less clutter from background noise. By applying Equation (10), the signal-to-noise ratio of the network input images is enhanced, making it easier to extract the polarization characteristics of the target.

### 2.3. Network Structure

Ba et. al. [16] proposed a method that combines the UNet architecture with a polarization 3D imaging technique (hereinafter referred to as SfP-UNet), and it demonstrated stable performance under different illumination conditions. However, the output normal maps had the issue of losing high-frequency detail area information. In this study, a network based on the U^2^Net architecture [32] is proposed for improved SfP performance (hereinafter referred to as SfP-U^2^Net), as shown in Figure 4. Compared to UNet [25], the proposed approach incorporates a residual U block (RSU) to replace single convolutional layer or deconvolutional layer operation, with the aim of addressing the problem of a narrow receptive field. Despite the gradual decrease in spatial size of the input image during the downsampling process, U^2^Net is able to extract more comprehensive polarization information, including both global and local context information. At the same time, U^2^Net also addresses the problem of increased computation caused by dilated convolution and obtains a larger receptive field without increasing computational cost. Additionally, compared to the UNet-based polarization 3D imaging method [23], the proposed method does not significantly increase the computational cost while increasing the depth of the entire architecture. As depicted in Figure 4, SfP-U^2^Net extracts multiscale features from downsampled feature maps and obtains high-resolution feature maps through progressive upsampling, concatenation, and convolutional encoding. This process reduces the loss of context information caused by direct upsampling on large scales.

The input of the network consisted of polarization images, $I_{rawS},I_{rawD}$, $\phi_{e}$ and $I_{\rho }$. Each decoder output 3-channel feature maps, resulting in a total of six three-channel feature maps, as illustrated in Figure 4. These six feature maps were upsampled to the same size and concatenated as the input of the Final Layer. The concatenated feature maps were fed into the network to generate an output normal image.

### 2.4. Loss Function

The loss function is expressed in [21] as:(11)$$L_{cosine}=\frac{\sum_{i}^{W}\sum_{j}^{H}\left(1-\;\langle {\hat{N}}_{i,j},\;N_{i,j}\rangle \right)}{W\times H}$$
where 〈·,·〉 denotes the dot product, ${\hat{N}}_{i,j}$ denotes the estimated surface normal at the pixel location $\left(i,j\right)$, $N_{i,j}$ denotes the corresponding ground truth surface normal, *W* and *H* represent the width and height of the image, respectively. Equation (11) is minimized when ${\hat{N}}_{i,j}$ and $N_{i,j}$ have the same orientation. However, Equation (11) also takes into account the influence of background pixels (the image contains two regions of the target and the background, we refer to the pixels in the background region as background pixels) in the image. To improve the accuracy of the output object surface normal and reduce the impact of background pixels, the loss function was modified to focus more on the normal estimation results by excluding the background pixels from the denominator and numerator of Equation (12). The improved loss function is defined as follows:(12)$$L=\frac{\sum_{i}^{W}\sum_{j}^{H}\left(1-\;\langle {\hat{N}}_{i,j},\;N_{i,j}\rangle \right)-m}{W\times H-m}$$
where *m* represents the number of background pixels.

### 2.5. Experimental Device

To evaluate the effectiveness of our proposed SfP-U^2^Net, experiments were conducted using the DeepSfP dataset [21], which is explained in Section 3.1. Furthermore, an experimental setup was created to capture polarization images of objects in various lighting conditions, both indoors and outdoors, as shown in Figure 5, to validate the proposed SfP-U^2^Net. The acquired polarization images were used for both demonstration and verification purposes. The objects photographed included resin models, plastic jars, and rubber balloons. Each object was photographed from three different directions: front, side, and back. Polarization images were captured using a FLIR Blackfly S USB3 polarization camera, which can capture four polarization images at angles of 0°, 45°, 90°, and 135° in a single shot. The resolution of the polarization imaging sensor was 2448 × 2048 pixels and the resolution of a single polarization image was 1224 × 1024 pixels. The camera was equipped with a 25 mm F2.8 lens. A nonpolarized light source (SCHOTT EasyLED spotlight) was used to illuminate the test object and a piece of a light-absorbing cloth was used as the imaging background. The experimental setup was used to collect outdoor polarization image of the testing objects under cloudy weather conditions. These collected polarization images were fed into U^2^Net [32] to obtain the mask required for the inference.

## 3. Data and Implementation Details

### 3.1. Dataset

The proposed SfP-U^2^Net was trained and verified using the DeepSfP dataset, which comprises polar images of 25 different objects. Each object had 4 different orientations (front, back, left, right) and a total of 100 object-orientation combinations. For each orientation, polar images were captured under three different lighting conditions (indoor, cloudy outdoor, and sunny outdoor). The DeepSfP dataset, comprising 300 polarization images, is illustrated in Figure 6 [21]. The polarization camera captured images at four different polarization orientations, namely, 0°, 45°, 90°, and 135°, in a single shot. The structured light-based 3D scanner obtained the high-quality 3D shape of the object, which was used to calculate the ground truth normal map. In addition, the polarization images collected using the experimental setup described in Section 2.5 were also used to verify the performance of the proposed method.

### 3.2. Training

SfP-U^2^Net was implemented using PyTorch and trained for 1000 epochs on six Nvidia Tesla A100 GPUs (80 GB of memory) with a batch size of 24. We used the Adam optimizer [33] with an initial learning rate of 1 × 10^−^³ and employed a cosine decay scheduler for the learning rate. The learning rate was also scaled linearly with the batch size. In previous studies such as [21], training set images were randomly cropped to account for the different sizes and positions of the target objects within the dataset. However, while random cropping can enhance data, it also has its drawbacks such as generating images with mostly background pixels, leading to low network performance. The latter occurs as some parts were not learned by the network due to their exclusion from the cropping range. To address this issue, we improved the random cropping technique by calculating the proportion of objects in the image after random cropping, ensuring that the proportion of object pixels in all network input images was greater than 50%.

### 3.3. Inference

To improve the inference efficiency, the 1224 × 1024 pixel input image was divided into 16 patches of size 256 × 256 pixels and then fed into the network. The 16 obtained surface normal image patches were then stitched together to form a 1024 × 1024 surface normal map. This process was repeated 32 times to obtain the mean values of the 32 surface normal images. To increase the robustness of the network, the input images were randomly shifted before being split into smaller patches. This ensures that the object in the input image is always present within the image and not shifted outside of it. By using random movement, the calculated mean normal map can better reflect the overall performance of the network.

## 4. Experimental Results

In this section, we evaluated the performance of the proposed SfP-U^2^Net using the test set from the DeepSfP dataset and the data collected by the self-built experimental setup described in Section 2.5. In this paper, a comparative analysis of the proposed method and previous UNet-based methods was presented. A series of ablation experiments were conducted on both SfP-U^2^Net and SfP-UNet frameworks to investigate the effectiveness of the proposed training method, inference method, loss function, and modified degree of polarization image in improving the network’s SfP performance. All the network models were trained on the training set from the DeepSfP dataset. The MAE was used as a quantitative measure to evaluate the network performance.

### 4.1. Network Input

To determine the optimal network input, we investigated the effect of different polarization representation inputs on network performance. Five groups of polarization representations were used as inputs for testing, as shown in Table 1. $I_{raw}$ represents a combination of $I_{rawS}$ and $I_{rawD}$. Input 1 used the training method, inference method, and loss function proposed in [21]. Inputs 2 to 6 used the training method, inference method, and loss function proposed in this study and described in Section 2 and Section 3. Input 3 was created by adding $I_{\rho }$ to Input 2, which was used to demonstrate the effectiveness of $I_{\rho }$ in improving SfP performance. Input 4 was the input for network proposed in [24]. Input 5 replaced ρ in Input 4 with $I_{\rho }$, which aimed to prove the effectiveness of $I_{\rho }$ in improving SfP performance. Comparing Inputs 6 and 5, the $I_{un}$ term was replaced with $I_{raw}$ to provide specular and diffuse information to the network.

### 4.2. Ablation Experiments

The DeepSfP dataset was used for the ablation experiments. Figure 7 shows the normal maps estimated using the analytical and deep learning-based SfP methods. Columns 4 to 6 depict the normal estimations obtained using the deep-learning-based SfP-UNet model [21]. However, the input polarization representation, training method, inference method, and loss function have been modified as described in Section 4.1. Compared with the analytical SfP method [34], the SfP-UNet model provides more accurate normal map. The MAE was also significantly reduced, as shown in the top-left corner of the sub-images. Additionally, even though the normal maps shown in Column 4–6 were all obtained using the UNet-based SfP method [21], the surface normal estimations were quite different. The experimental results demonstrate that modifying the polarization representation input, training method, inference method, and loss function can improve SfP performance.

As shown in Figure 7, two different inputs were tested for the SfP-UNet framework. By comparing the normal maps obtained by Inputs 1 and 2, it can be concluded that the network performance can be improved by adjusting the training method, inference method, and loss function. As shown in Columns 4 and 5 in Figure 7, the surface normal maps obtained using Input 2 were better. The use of Input 2 resulted in a significantly lower MAE for Christmas and Flamingo. However, there was information loss during reconstruction in the fine-textured areas of the target, resulting in inaccurate normal estimation in those regions and decreasing the overall reconstruction accuracy. This was particularly evident in the legs of the dragon and the back of the horse. Hence, $I_{\rho }$ was introduced as the network input to enhance the contrast between the target and background polarization in the input image, decrease background noise, and facilitate the network to extract more targeted polarization features from the input.

As shown in the 2nd and 3rd row of Table 2, and columns 5 and 6 in Figure 7, adding $I_{\rho }$ to the network input reduces the MAE. The issue of texture information loss still persisted despite the proposed improvements. It is our belief that by enabling the network to extract more comprehensive information from the network layer, it may be feasible to enhance the accuracy of normal estimation without compromising object texture information. Therefore, the U^2^Net network was introduced to address this problem.

### 4.3. SfP-U^2^Net

We propose the use of SfP-U^2^Net to improve SfP performance by preserving the detailed information of the object while accurately estimating surface normals. In addition, in order to reduce the computational cost, we tested various forms of network input to improve network performance while minimizing resource consumption.

The most challenging aspect of the DeepSfP dataset is the accurate 3D reconstruction of objects with complex shapes and textures, such as Horse and Dragon. These objects are known to have intricate details and specular reflections, which makes the SfP task particularly challenging. The SfP-U^2^Net model’s ability to extract texture information and estimate surface normals was evaluated by capturing the normal map of the fine-textured region (red box) in Figure 8. Performance was assessed using median angular error and the percentage of pixels falling within 11.25°, 22.5°, and 30.0° error ranges. Compared to the SfP-UNet model, the SfP-U^2^Net model produced more precise and comprehensive reconstructions of the objects in question by demonstrating superior ability to extract texture information and estimate surface normals under the same network input.

As shown in Table 3, different types of network inputs were tested using the SfP-U^2^Net. The results indicate that under the same input conditions, SfP-U^2^Net not only reduces memory consumption but also obtains a lower mean angle error, resulting in a significant improvement in network performance. Table 3 shows that the MAE obtained by SfP-U^2^Net could be decreased to 18.55° when using Input 2 as the network input, which was lower than that of the UNet-SfP method. We further tested inputs 3, 4, 5, and 6 to identify the best polarization representation as the network input. From the results of input 3, $I_{\rho }$ could indeed improve the accuracy of the network to estimate the surface normal of the object. ρ, $\phi_{e}$, and $I_{un}$ replaced the estimated normal in Input 4. Compared to Input 2, the MAE of Input 4 was decreased by 0.09°, while also significantly reducing the computation time as the estimated surface normal includes a total of nine image channels, whereas ρ, $\phi_{e}$, and $I_{un}$ only required four image channels for representation. By comparing Inputs 4 and 5, it was observed that adding $I_{\rho }$ further improved the performance of the network. It can be concluded that $I_{raw}$ was more suitable for U^2^Net-SfP to extract polarization information than $I_{un}$ by comparing Inputs 5 and 6.

As shown in Figure 9, the proposed U^2^Net-SfP method demonstrates excellent performance on objects such as Box, Christmas, Horse (Front), Vase, Flamingo, and Boll, and performs well on Horse (Right) and Dragon as well. The use of random offsetting of the input image before inference allows the model to achieve optimal SfP performance for specific objects under certain lighting conditions. For example, the MAE of Horse (front) was reduced to 6.02°. The replacement of $I_{un}$ with $I_{raw}$ in the proposed U^2^Net-SfP method improved SfP results by providing more accurate clues for estimating the surface normal of the object. The separation of specular and diffuse reflection components in this study was based on approximate estimations using the equations described in Section 2. However, future research could focus on developing accurate methods for separating specular and diffuse reflection components to achieve even more accurate restoration of target surface normal maps. Moreover, the replacement of the estimated normal with $I_{raw}$, after separation of specular and diffuse reflection, significantly reduced the calculation time by minimizing the amount of input data. As seen in Table 3, the calculation time of Input 6–7 was significantly reduced as a result.

We also compared our method with previous methods. Table 4 shows a comparison of the proposed U^2^Net-SfP method with other previously reported SfP methods using the DeepSfP dataset. When Compared to three previously reported analytical SfP methods (Miyazaki [35], Mahmoud [10], and Smith [34]) and two deep learning-based SfP methods (DeepSfP [21] and SPW [24]), the proposed method achieves the lowest MAE metric.

### 4.4. Experimental Results of the Actual Shooting

We evaluated the performance of our network model using the data collected by the experimental setup described in Section 2.5. As shown in Figure 10 and Figure 11, the texture information of the objects was well preserved in the surface normal maps obtained under indoor illumination conditions. Additionally, the model was tested on polarized images captured under more complex lighting conditions such as on a cloudy day. Although the SfP performance of the model was still robust under these conditions, some detailed texture information on the surface normal maps was not fully reconstructed. As shown in Figure 11 A2-Normal, the texture information of the sculpture’s face was not accurately captured.

## 5. Conclusions

The U^2^Net architecture has been introduced to address the limitations of the UNet-based network in recovering target texture details and improving surface normal estimation accuracy. Quantitative and qualitative analysis through a public dataset have confirmed that the proposed method can retain more texture details while accurately restoring the surface normal direction. Furthermore, by using the Stokes-vector-based parameter and the extracted specular and diffuse reflection components, the representation of the physical prior input was improved and the computation time was reduced. These modifications led to a better representation of the object’s polarization characteristics and improved the accuracy of the surface normal restoration. Additionally, our trained model had fewer parameters resulting in reduced computational costs. However, the attention given to texture features by the network played a significant role in surface normal estimation and further improvements could be made to enhance the feature extraction capabilities of the proposed method. The future work will focus on exploring more accurate separation of specular and diffuse reflection components using polarization representation and deep-learning methods with the goal of achieving accurate estimation of the full-frame surface normal using a single network model and original polarization images.

## Figures and Tables

**Figure 1 sensors-23-04592-f001:**
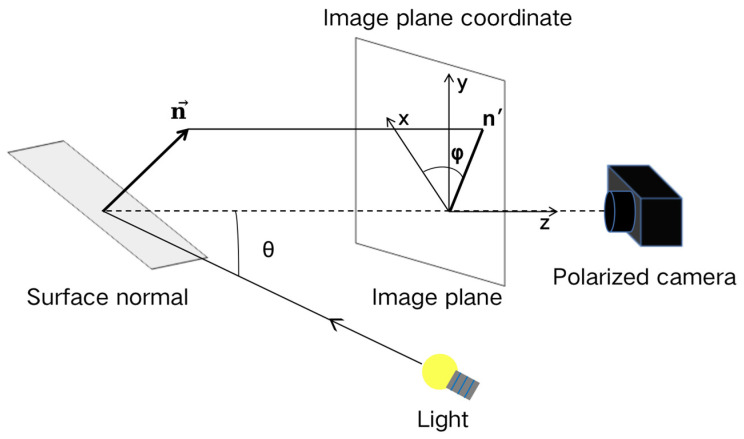
Schematic of the geometric relationship of the reflected light on the object surface.

**Figure 2 sensors-23-04592-f002:**
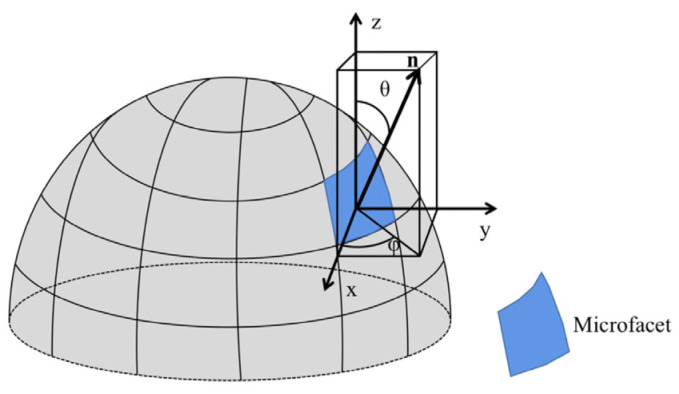
Schematic diagram of the microfacet normal vector.

**Figure 3 sensors-23-04592-f003:**
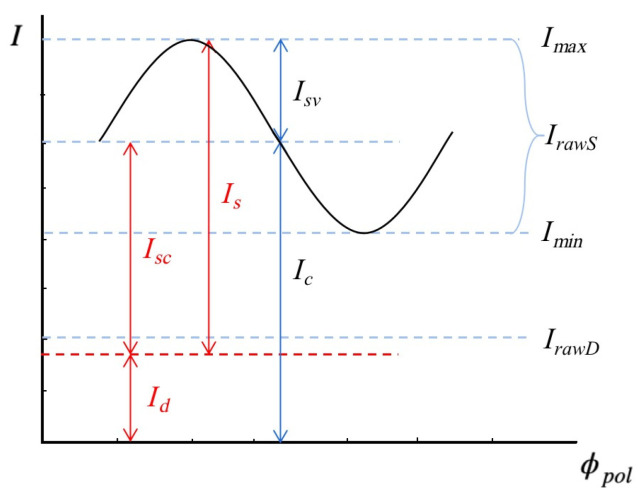
Relationship between image brightness and polarizer rotation angle. $I_{max}$ and $I_{min}$ represent the upper and lower bounds of measured light intensity I, respectively. $I_{rawS}$ and $I_{rawD}$ represent the approximate specular reflection (raw specular image) and diffuse reflection components (raw diffuse image), respectively. $I_{s}$ and $I_{d}$ represent specular reflection component and diffuse reflection component, respectively. $I_{c}$ denote the constant reflection component, which is the offset of $I_{rawS}$, or the sum of diffuse component $I_{d}$ and unpolarized specular component $I_{sc}$. For different polarization orientation $\phi_{pol}$, the specular reflection component can be wrote as the sum of $I_{sc}$ and a cosine function term with amplitude $I_{sv}$. [29].

**Figure 4 sensors-23-04592-f004:**
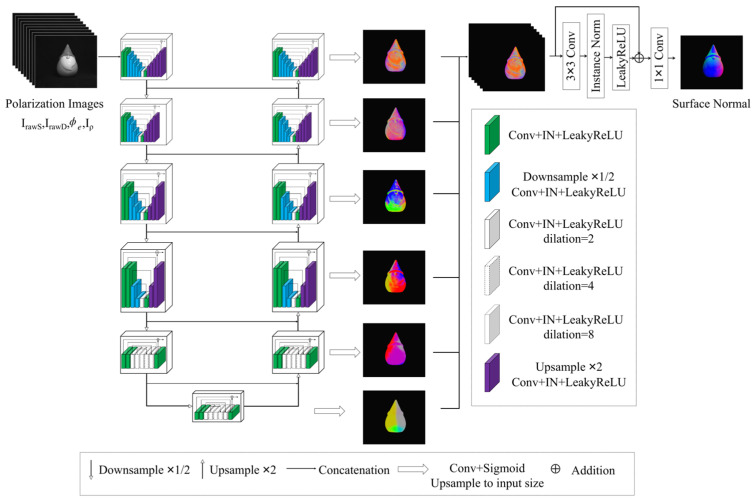
Overview of the proposed method. The input of the proposed network consists of two parts: (1) Polarization images $I_{0}$, $I_{45}$, $I_{90}$, $I_{135}$. (2) Specular reflection component $I_{rawS}$, diffuse reflection component $I_{rawD}$, encoded phase angle $\phi_{e}$ and Stokes-vector-based parameter $I_{\rho }$ calculated from polarization images. The concatenated polarization imaging inputs are fed into the neural network to estimate normal image.

**Figure 5 sensors-23-04592-f005:**
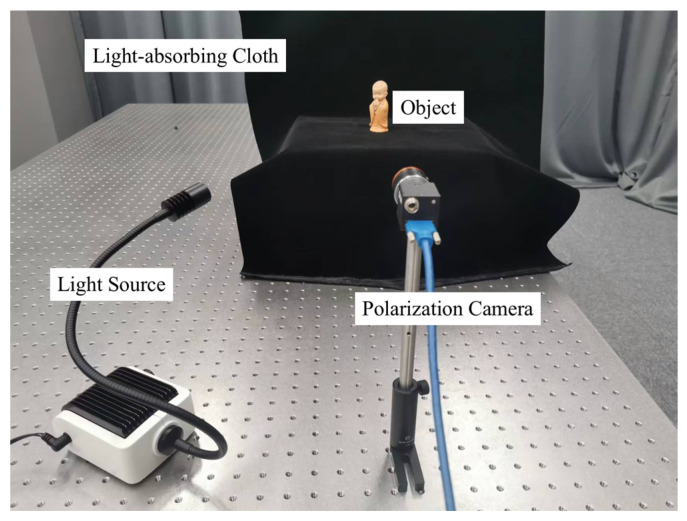
Experimental setup for data collection.

**Figure 6 sensors-23-04592-f006:**
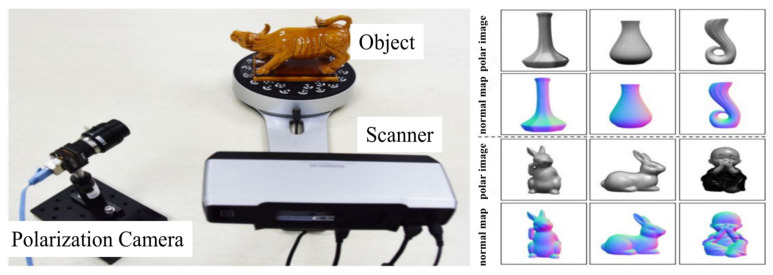
DeepSfP dataset created by using a polarization camera to capture four polarization images of an object and a scanner to obtain the 3D shape of the object. Polarization images are shown with a polarization angle of 0°. The corresponding normal maps are aligned below.

**Figure 7 sensors-23-04592-f007:**
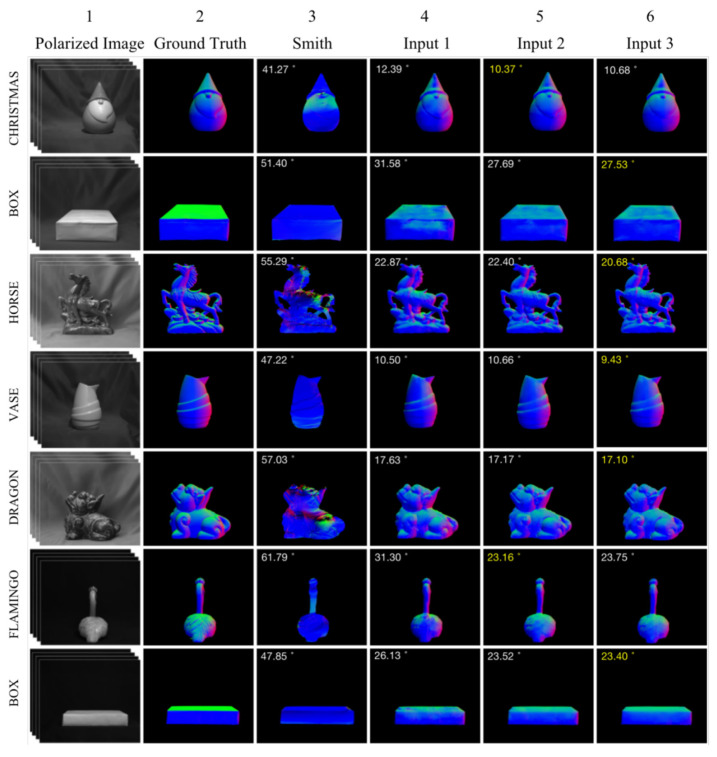
Normal maps calculated using the analytical SfP method [34] and the deep learning methods. The angles appearing at the top left of each panel are MAEs. Columns 4–6 are the normal maps estimated using the SfP-UNet model [21] with different loss function and model input.

**Figure 8 sensors-23-04592-f008:**
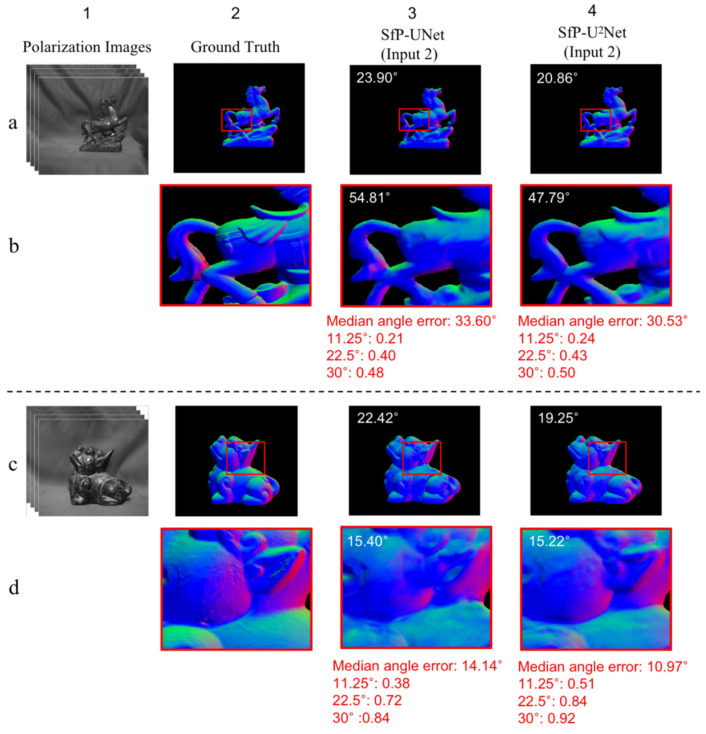
Surface normal maps obtained with SfP-U^2^Net and SfP-UNet are presented, with MAEs shown at the top left corner of each image. Rows (**a**,**c**) estimate the surface normals of horse and dragon using the SfP-UNet and SfP-U2Net. Rows (**b**,**d**) display an enlarged view of the red boxes in the normal maps. Apart from MAE, we also assessed the enlarged regions in terms of median angular error and the proportion of pixels whose angles fall within 11.25°, 22.5°, and 30.0° error ranges.

**Figure 9 sensors-23-04592-f009:**
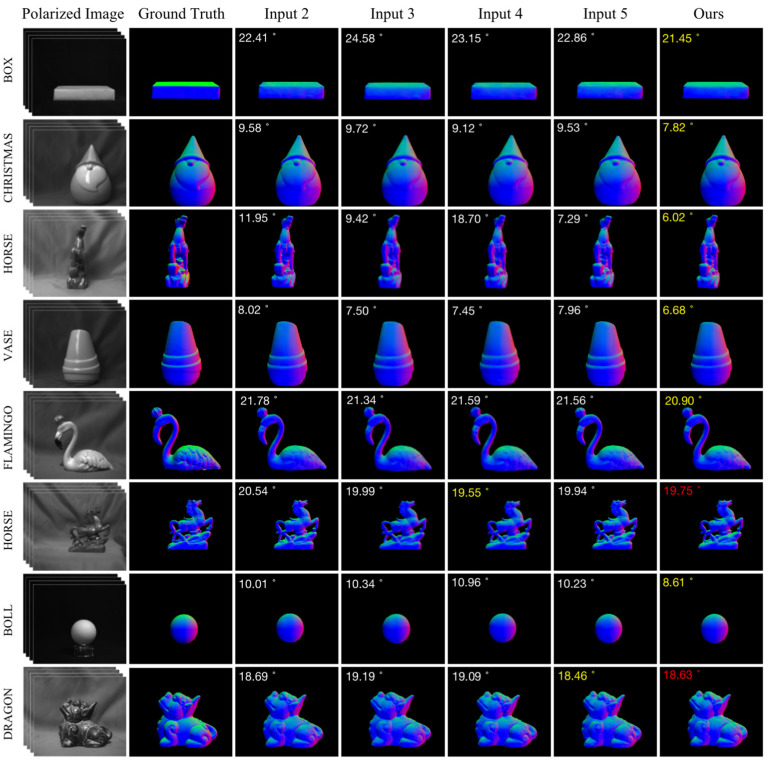
Verification of the proposed SfP-U^2^Net using DeepSfP dataset, different kinds of network inputs are tested. The angles appearing at the top left of each panel are MAEs. The best MAEs are highlighted in yellow. The next best MAE is highlighted in red font.

**Figure 10 sensors-23-04592-f010:**
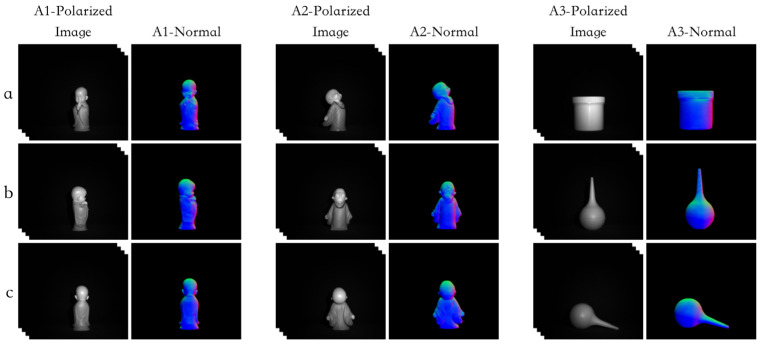
For polarization images taken under indoor illumination conditions, the surface normal estimation calculated using the proposed SfP-U^2^Net. Rows (**a**–**c**) show the surface normal estimations of different objects in different view. The detail features and context information were well reconstructed.

**Figure 11 sensors-23-04592-f011:**
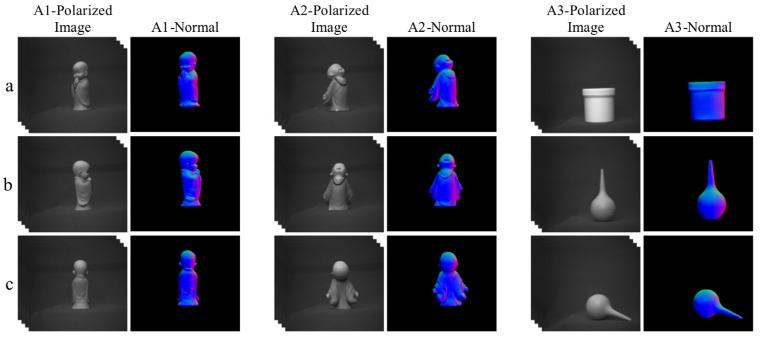
For polarization images taken in a cloudy day under outdoor illumination conditions, the surface normal estimation was calculated using the proposed SfP-U^2^Net. Rows (**a**–**c**) show the surface normal estimations of different objects in different view. The detail features and context information were well reconstructed.

**Table 1 sensors-23-04592-t001:** Performance of the SfP methods are tested under different network inputs.

	Polarization Images	Estimated Normal	$I_{raw}$	$I_{\rho }$	ρ	$\phi_{e}$	$I_{un}$
Input 1 [19]	√	√					
Input 2	√	√					
Input 3	√	√		√			
Input 4 [22]	√				√	√	√
Input 5	√			√		√	√
Input 6 (Ours)	√		√	√		√	

**Table 2 sensors-23-04592-t002:** The performance of SfP-UNet is evaluated on the DeepSfP dataset using three input sets, as described in Table 1. The accuracy of surface normal estimation by the network is evaluated using mean angular error. **↓** indicates that the lower the mean angular error, the better the network performance.

Network	Input		Mean Angular Error ↓	Parameters (M)	Time (s)
SfP-UNet	1	Input 1	21.76°	49.59	0.545
	2	Input 2	20.15°	49.59	0.545
	3	Input 3	19.88°	49.59	0.550

**Table 3 sensors-23-04592-t003:** DeepSfP dataset to verify the performance of SfP-U^2^Net and SfP-UNet with different inputs. **↓** indicates that the lower the mean angular error, the better the network performance.

Network	Input		Mean Angular Error ↓	Parameters (M)	Time (s)
SfP-UNet	1	Input 1	21.76°	49.59	0.545
	2	Input 2	20.15°	49.59	0.545
	3	Input 3	19.88°	49.59	0.550
SfP-U^2^Net	4	Input 2	18.55°	44.02	0.545
	5	Input 3	18.52°	44.02	0.550
	6	Input 4	18.46°	44.01	0.180
	7	Input 5	18.45°	44.01	0.180
	8	Input 6 (Ours)	17.60°	44.02	0.207

**Table 4 sensors-23-04592-t004:** The comparison of the proposed SfP-U^2^Net and previously reported SfP methods using DeepSfP dataset. Our method achieves the best score. **↓** indicates that the lower the mean angular error, the better the network performance.

Method	Mean Angular Error ↓
Miyazaki [35]	50.97
Mahmoud [10]	58.43
Smith [34]	51.84
DeepSfP [21]	21.76
SPW [24]	21.75
Ours	17.60

## Data Availability

Not applicable.

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
