# Peer review of "Three Dimensional Shape Reconstruction via Polarization Imaging and Deep Learning"

_sensors, 2023, doi:10.3390/s23104592_

Round 1

Author Response

Dear Reviewer,

Thank you for your constructive feedback on our paper. We have thoroughly revised the manuscript according to your suggestions and comments. Please find the point-by-point response attached to this email.

Thank you very much for your time and efforts in reviewing our paper. 

Best regards,

Author Response

(The authors gave the same response as above.)

Reviewer 3 Report

The paper presents the three dimensional shape reconstruction via polarization imaging and deep learning. The paper is interesting, however some matters need to be addressed before publication.

1. A better explanation of your contribution needs to be added in the Introduction Section. Please set differences of your work with respect to previous work. Please highligh the advantages, benefits and contributions of the new tecnique and results.

2. Provide more details of the methodology used.

3. There is a lot of papers dealing with this topic. Please cite references dealing with this topic.

4. It is needed a better description of the results. Please highligh the benefits, the advantages and contributions of your proposal.

Author Response

(The authors gave the same response as above.)

Reviewer 4 Report

This paper addresses these limitations like  current methods have limitations in restoring target texture details and surface normal estimation accuracy by proposing a novel polarization 3D imaging method based on the  architecture. Additionally, to enhance performance, the proposed networks polarization representation input is optimized by utilizing the modified degree of polarization image, as well as separated specular and diffuse reflection components. This reduces background noise impact, extracts more target polarization features, and provides more accurate cues for surface normal restoration. The experimental results show that the proposed model can provide more accurate surface normal estimates, and compared to the UNet architecture-based method, significantly im-proves mean angle error, calculation time, and model size. Deep learning-based polarization 3D imaging approaches, which train networks in a data-driven manner, can estimate the surface normal distribution of a target under passive lighting conditions. In order to improve this work, the following questions can be considered and the manuscript need to be revised accordingly.

(1) The authors are suggested to compare and discussion some recent works in the ultrafast photonics fields, (i.e. Ultrafast Science, 2022, 9767251, 16, 2022; Ultrafast Science, 2022, 9870325, 6, 2022; Ultrafast Science, 3, 0006, 2023; Phys. Rev. Lett., 121, 023905 2018; Laser Photon. Rev. 13, 1800333, 2019; Phys. Rev. Lett., 123, 093901, 2019).

(2) In this paper, we can analyze whether binocular stereoscopic method can be used to solve the problem of zenith Angle and azimuth Angle.

(3) In this paper, it is mentioned that the method to eliminate the multi-value problem of incident Angle and azimuth Angle based on specular reflection has many problems such as multiple detection and complex solution. It is suggested to introduce it in detail again, so as to make a clear contrast introduction to deep learning methods.

(4) Please pay attention to the layout of the pictures in the following article, which can make the article more beautiful.

(5) It is mentioned in the article that the  method can obtain high-resolution feature maps through progressive up-sampling, cascade, convolution and coding. May I ask whether this step is realized by increasing the number of layers of neural network? Could you give us more details?

(6) In order to optimize the  model, it is recommended to further study the principle and solution of the incomplete reconstruction of detailed texture information on the overcast surface normal map.

(7) Some related work with more in-depth discussion should be mentioned in the manuscript. (i.e. Computer Science 1671-3133(2007)); Optics & Laser Technology, 146, 107546, (2022); Transportation 1000-680X(2003)25; ACS Photonics, 7, 9, 2440-2447, 2020; Optics & Laser Technology, 151, 108016, (2022).

Author Response

(The authors gave the same response as above.)

Reviewer 5 Report

This work proposes a deep learning-based polarization 3D imaging method to estimate the surface normal distribution of a target under passive lighting conditions. Based on the U2Net architecture, this method improves the quality of restored target texture and the accuracy of surface normal estimation. In data preprocessing, a random cropping technique is adopted, which reduces background noise, extracts more target polarization features, and provides more accurate cues for surface normal restoration. The experimental results are compared with that of the SfP-UNet. The comparison of the proposed SfP-U2Net and previously reported SfP methods using DeepSfP dataset is presented as well. Here are some concerns:

1.      More analysis on the performance in experiment is required to show the advantage of the proposed method, such as adding evaluation indicators.

2.      Some of the references used in Table 4 for comparison should be replaced by state-of-the-art research results.

Author Response

(The authors gave the same response as above.)

Reviewer 6 Report

See attached file

Author Response

(The authors gave the same response as above.)

Round 2

Reviewer 2 Report

The paper can now be accepted.